# Genotyping and Antimicrobial Susceptibility Profiling of *Streptococcus uberis* Isolated from a Clinical Bovine Mastitis Outbreak in a Dairy Farm

**DOI:** 10.3390/antibiotics10060644

**Published:** 2021-05-28

**Authors:** Valentina Monistero, Antonio Barberio, Paola Cremonesi, Bianca Castiglioni, Stefano Morandi, Desiree C. K. Lassen, Lærke B. Astrup, Clara Locatelli, Renata Piccinini, M. Filippa Addis, Valerio Bronzo, Paolo Moroni

**Affiliations:** 1Dipartimento di Medicina Veterinaria, Università degli Studi di Milano, 26900 Lodi, Italy; valentina.monistero@unimi.it (V.M.); clara.locatelli@unimi.it (C.L.); renata.piccinini@unimi.it (R.P.); filippa.addis@unimi.it (M.F.A.); valerio.bronzo@unimi.it (V.B.); paolo.moroni@unimi.it (P.M.); 2Istituto Zooprofilattico Sperimentale delle Venezie, 35020 Legnaro, Italy; abarberio@izsvenezie.it; 3Institute of Agricultural Biology and Biotechnology, National Research Council, 26900 Lodi, Italy; bianca.castiglioni@ibba.cnr.it; 4Institute of Sciences of Food Production, Italian National Research Council, 20133 Milan, Italy; stefano.morandi@ispa.cnr.it; 5Centre for Diagnostics, DTU Health Tech, Technical University of Denmark, 2800 Kongens Lyngby, Denmark; dcokla@dtu.dk (D.C.K.L.); lboast@vet.dtu.dk (L.B.A.); 6Quality Milk Production Services, Animal Health Diagnostic Center, Cornell University, Ithaca, NY 14850, USA

**Keywords:** *Streptococcus uberis*, cow, clinical mastitis, RAPD, MIC

## Abstract

*Streptococcus uberis*, an environmental pathogen responsible also for contagious transmission, has been increasingly implicated in clinical mastitis (CM) cases in Europe. We described a 4-month epidemiological investigation of *Strep. uberis* CM cases in an Italian dairy farm. We determined molecular characteristics and phenotypic antimicrobial resistance of 71 *Strep. uberis* isolates from dairy cows with CM. Genotypic variability was investigated via multiplex PCR of housekeeping and virulence genes, and by RAPD-PCR typing. Antimicrobial susceptibility was assessed for 14 antimicrobials by MIC assay. All the isolates carried the 11 genes investigated. At 90% similarity, two distinct clusters, grouping 69 of the 71 isolates, were detected in the dendrogram derived from the primer ERIC1. The predominant cluster I could be separated into two subclusters, containing 38 and 14 isolates, respectively. *Strep. uberis* strains belonging to the same RAPD pattern differed in their resistance profiles. Most (97.2%) of them were resistant to at least one of the drugs tested, but only 25.4% showed a multidrug resistance phenotype. The highest resistance rate was observed for lincomycin (93%), followed by tetracycline (85.9%). This study confirmed a low prevalence of β-lactam resistance in *Strep. uberis,* with only one isolate showing resistance to six antimicrobial classes, including cephalosporins.

## 1. Introduction

*Streptococcus uberis* is primarily classified as an environmental pathogen causing about one-third of all intramammary infection (IMI) cases in lactating and nonlactating cows worldwide [1,2]. Recently, an increased prevalence of *Strep. uberis* isolates associated with clinical mastitis (CM) has been reported in several European countries [3,4].

Although its primary reservoir is the dairy farm environment [5,6,7], some *Strep. uberis* strains show a contagious transmission mode [8,9], deserving further investigation. Molecular genotyping contributes to understanding the modes of dissemination and can be useful in epidemiological studies of *Strep. uberis* mastitis. The most commonly used subtyping methods are MLVA (Multiple Loci VNTR Analysis; VNTR, Variable Number of Tandem Repeats), MLST (MultiLocus Sequencing typing), PFGE (Pulsed Field Gel Electrophoresis), and RAPD-PCR (Random Amplified Polymorphic DNA-PCR) [8,10,11,12]. Generally, *Strep. uberis* has a nonclonal population structure but clonal strains in different cows have been detected [13]. Zadoks and collaborators [14] applied RAPD-PCR to demonstrate the genetic homogeneity of *Strep. uberis* within some farms. Recently, this latter technique was shown to be reliable in typing *Strep. uberis* and identifying clonal strains in different cows within herds [15]. Despite bacterial diversity, most *Strep. uberis* strains share the same combination of highly conserved virulence genes (*hasA, hasB, hasC, oppF, pauA, sua)* [16,17,18], strongly associated with the pathogenesis of IMI. The two multiplex PCR (mPCR) assays developed by Calonzi and collaborators [19] allow the detection of these genes for a rapid characterization of *Strep. uberis* strains. Additionally, the *lbp*-producing iron-binding protein aids in adherence to and internalization into bovine mammary epithelial cells [20].

Antimicrobial therapy is still the primary strategy to control and treat bovine IMI. The widespread and sometimes inappropriate use of drugs to cure clinical cases has led to an increasing diffusion of antimicrobial resistance (AMR) in Streptococci, resulting in health problems at the herd level [21]. Although β-lactam antibiotics, especially penicillin and cephalosporins, are frequently recommended to treat clinical IMI [22], an increased ability of the bacterium to develop resistance to cephalosporins has been reported [15]. Recent reports have also shown the resistance of *Strep. uberis* to macrolides, lincosamides, and tetracyclines [11,23,24].

We investigated the epidemiology of *Strep. uberis* CM cases in a dairy farm during a 4-month study period. We determined the genotypes, the virulence profiles and the AMR patterns of 71 *Strep. uberis* strains in order to understand their heterogeneity and the mode of transmission within the herd.

## 2. Results

### 2.1. Virulence Profiling

The four housekeeping genes investigated (*cpn60, gapC sodA* and *tuf*) and the combination of the seven virulence-associated genes (*lbp, hasA, hasB, hasC, oppF, pauA, sua*) were detected in 100% of the analyzed isolates.

### 2.2. Genotyping

The 71 *Strep. uberis* isolates were characterized by RAPD-PCR analysis with primer ERIC1. As isolates with a similarity coefficient equal to or higher than 90% can be considered as closely related, the amplification profiles identified clonal strains in different cows and displayed conserved patterns within the herd. At 90% similarity, two distinct clusters were detected in the dendrogram derived from ERIC1 (Figure 1). These two major clusters, indicated with roman numerals (I and II), grouped 69 of the 71 *Strep. uberis* isolates. A unique cluster (cluster I) was predominant among the herd isolates (73.3%) and could be separated into two subclusters (Ia and Ib), containing 38 and 14 isolates, respectively. Cluster II included 17 (23.9%) isolates, while the remaining two (2.8%) isolates did not enter any cluster.

### 2.3. Antimicrobial Profiling

The phenotypic results obtained by MIC assay (Table 1) revealed that the 71 *Strep. uberis* isolates had different degrees of resistance to the 14 active substances and most (97.2%) were not inhibited by at least one of the drugs tested, showing the highest resistance rate to lincomycin (93%) and tetracycline (85.9%). Analyzing the resistance to multiple classes of antimicrobials, 14.1% of the *Strep. uberis* isolates were resistant to one antimicrobial class, but most were resistant to two (57.7%), or three and more classes (25.4%). In particular, the percentage of resistance to four and five antimicrobial classes was 7.1% and 2.8% respectively, and only one isolate was classified as resistant to six different classes. Table 1 reports the distribution of the MIC inhibiting the growth of 50% and 90% of the isolates (MIC_50_ and MIC_90_) for all the antimicrobials tested, and shows the highest inhibiting concentrations for lincomycin and tetracycline, with MIC_90_ values even higher than the resistance breakpoints. The lowest inhibiting concentrations were for erythromycin and most of the β-lactams. Low levels of resistance (8.5%) were found to erythromycin, and also to fluoroquinolones (enrofloxacin). Resistance to β-lactam antibiotics was present, with 11 (15.5%) isolates resistant to penicillin, three (4.2%) to oxacillin, one to first-generation (cefazolin) and fourth-generation (cefquinome) cephalosporins, and two isolates resistant to third-generation (cefoperazone and ceftiofur). 

### 2.4. Distribution of Antimicrobial Resistance between Genotypic Clusters

Table 2 shows the differences in antimicrobial resistance observed between the two clusters obtained by the RAPD-PCR analysis. The distribution of MIC_50_ and the MIC_90_ values for florfenicol, lincomycin and tetracycline was identical throughout the RAPD clusters, while there were differences for the other antimicrobials tested. The cluster I isolates showed significantly higher MIC values than those of cluster II for amoxicillin plus clavulanate (*p* value = 0.034) and for ceftiofur (*p* value = 0.01) (Appendix A).

## 3. Discussion

A 4-month retrospective cohort study was undertaken to characterize 71 *Strep. uberis* isolates obtained from CM cases at an Italian dairy farm. The genetic analyses revealed that *Strep. uberis* mastitis cases in this herd could be linked to a relatively restricted number of cow-adapted strains, grouped in the same genotypic cluster. In the present study, the genetic similarity of *Strep. uberis* strains analyzed was evident in the PCR analysis, revealing that their virulence profile was characterized by a particular combination of genes that may be responsible for the high incidence of CM cases. The genes *sua, lbp* and *pauA* could express potential adherence determinants enhancing the ability to cause clinical disease [17,30]. The high prevalence of such genes was in agreement with a previous study [31], demonstrating that they may confer *Strep. uberis* an advantage in survival and colonization in nutritionally limited environments [32,33]. The *oppF* is similarly important during bacterial growth in milk [34] because of its role in the acquisition and utilization of essential amino acids from milk. The *has* operon (*hasABC*), conferring resistance to phagocytosis, may play a role in the development of IMI [35], although not strictly required in the onset of infection [36]. 

The genetic similarity of *Strep. uberis* strains within the herd was confirmed by RAPD-PCR, a rapid and inexpensive tool to discriminate individual strains of *Strep. uberis* [15]. In our study, RAPD-PCR analysis identified closely related *Strep. uberis* strains in different cows. The use of the primer ERIC1 detected two major clusters (I and II) and grouped most (73.3%) of the strains in cluster I, similar to what was reported in a previous paper [14]. Despite the high level of genetic relatedness, the remaining 26.7% of *Strep. uberis* strains were grouped in a minor genotypic cluster or did not belong to any cluster. 

The clear prevalence of a single RAPD cluster might suggest the predominance of a contagious behavior [12], although an environmental transmission should not be ruled out [8]. The minor cluster might group the strains coming from extramammary sites [11], including manure and bedding materials [6,7], with less opportunities to be transmitted from cow to cow. Overall, the spreading of *Strep. uberis* infections might originate from the exposure of the teat to the bovine reservoir, as well as to a common environmental source, because of the ability of *Strep. uberis* to persist in various spots of the dairy environment [5]. The coexistence of these two modes of transmission might be the reason for the high number of *Strep. uberis* CM cases within the studied herd, as recently suggested [9]. Control measures focused on the improvement of environmental hygiene [37] may not be sufficient and different management decisions can be recommended on each farm [8,9]. Accordingly, the identification of strains likely to be responsible for cow-to-cow spread within a herd might help with the prevention strategies for the control of *Strep. uberis* mastitis [38]. 

The *Strep. uberis* strains belonging to the same RAPD pattern differed in their resistance profiles, as previously found by Tomazi and colleagues [15]. 

The most common (81.7%) resistance phenotype was for lincosamides (lincomycin) and tetracyclines. In line with previous findings [39], we observed a high (85.9%) resistance rate and MIC_90_ of *Strep. uberis* for tetracycline, although formulations containing tetracycline are not used to treat CM in Italy, but rather for other cattle diseases and in different food-production animals [22,40]. The resulting increase of tetracycline resistance in *Strep. uberis* from IMI might reflect the environmental nature of this pathogen. Here, most isolates (93%) showed high MIC_90_ values and resistance to lincosamides, in agreement with other European data [41,42]. The lincosamide resistance mechanism can be due to mobile genetic elements (MGEs). Among them, multiple integrative and conjugative elements were identified as carriers of resistance determinants, leading to cross-resistance to lincosamides and macrolides [43,44]. Of the *Strep. uberis* isolates resistant to lincosamides, only six were resistant also to erythromycin, a representative of macrolides. These results, similarly to those reported in Poland [45], showed that erythromycin resistance rate was lower than in other European countries [23,46]. Our data suggest that macrolides, the third-line antimicrobials recommended for the treatment of *Strep. uberis* infections [24], can be used in the herd, as most of the *Strep. uberis* strains displayed an L-phenotype (phenotypic resistance to lincosamides coupled with sensitivity to macrolides), according to Haenni and collaborators [43]. 

Overall, we found 25.4% of the *Strep. uberis* isolates had a multidrug resistance phenotype. On the contrary, Tian and collaborators [39] by using the Kirby Bauer disc diffusion test, reported a higher percentage (100%) of multiresistant streptococci in China. These discrepancies in antimicrobial resistance among countries could be due to the treatment of bovine IMI with different antimicrobials in diverse geographical areas [15], but also to the use of diverse susceptibility tests and interpretive criteria for phenotypic results [47]. It is difficult to accurately assess the level of antimicrobial resistance in mastitis pathogens using only clinical cut-off values, often adopted from other animal species, other groups of bacteria, or human medicine standards [45,47,48]. For this reason, the comparison of the MIC_50_ and MIC_90_ could provide useful information about the level of resistance, and it should always be performed when antimicrobial susceptibility and resistance data are analyzed. In our study, a good example of that is provided by the results of penicillin. This antimicrobial, considered as the most effective drug for treatment of *Strep. uberis* IMI [24,49], showed the highest resistance rate (15.5%) among β-lactams, but it had the same MIC_90_ of ampicillin and amoxicillin, which displayed a much lower resistance rate (1.4%). A misclassification of intermediate isolates to penicillin, due to the clinical breakpoints adopted, could probably be the source of this difference between the rate of resistance and the value of MIC_90_ for these antimicrobials, which share the same mechanisms of antimicrobial resistance against Streptococci. 

We found low levels of resistance also to cephalosporins, with only one *Strep. uberis* isolate displaying the widest resistance profile (six antimicrobial classes). This pattern of resistance to all β-lactams, including cephalosporins, could be associated with amino-acid substitutions in one or more of the penicillin-binding proteins (PBPs) encoded by *pbp* genes, whose variants might reduce the clinical efficacy of different β-lactams [50].

## 4. Materials and Methods

### 4.1. Herd

The herd is located in Lombardy (Northern Italy) and during the study it consisted of approximately 1400 lactating Holstein Friesian cows housed in free stalls, with an average daily milk production of 36 kg/cow and a bulk tank SCC of 268,000 cells/mL. Bedding in all lactating groups was chopped straw except in the groups of fresh and dry cows, where fresh-cut straw and rice husks were used. The feed consisted of a total mixed ration (TMR) of corn silage, alfalfa haylage, wheat straw, corn grain and protein mix. Cows were milked 3 times daily in 3 different parallel parlors (13 + 13). The farm is not under a DHIA (Dairy Herd Improvement Association) control program.

### 4.2. Sample Collection

From November of 2019 to February of 2020, a total of 422 milk samples were recovered from a quarter of clinical cases. Milk samples were collected by farm personnel trained to detect CM based on visibly abnormal milk (color, clots, blood) or udder changes (redness, heat, swelling, or pain) [51,52]. After disinfection of teat ends and discarding the first streams of fore-milk, milk was collected in 10 mL sterile vials labeled with the cow number and quarter. Milk samples were immediately stored at −20 °C and were shipped weekly to the Dipartimento di Medicina Veterinaria (DiMeVet) at the University of Milan. 

### 4.3. Bacteriological Culture and MALDI-TOF Confirmation

For bacteriological analysis, 100 µl of milk was plated onto blood agar plates containing 5% defibrinated sheep blood (Microbiol, Cagliari, Italy). Plates were incubated aerobically at 37 °C and evaluated after 24 and 48 h. Bacteria were identified according to the National Mastitis Council guidelines [53]. Catalase-negative and Gram-positive cocci were identified as Streptococci, and species were differentiated by further biochemical tests (growth in 6.5% NaCl broth, esculin hydrolysis, fermentation of ribose, sorbitol, sucrose, and inulin). A total of 76 isolates from quarter milk samples collected at the first CM case from 76 different cows were identified as *Strep. uberis*.

Matrix-assisted laser desorption ionization–time of flight mass spectrometry (MALDI-TOF MS) was used to confirm the identification of isolates as *Strep. uberis* at the DiMeVet microbiology laboratory. Isolates were freshly cultured on blood agar plates and cell material from an isolated colony was deposited on the target plate using a toothpick. Samples were overlaid with 1 μL of α-cyano-4-hydroxycinnamic acid in 50% acetonitrile with 2.5% trifluoroacetic acid (Bruker Daltonik GmbH, Bremen, Germany). The spectra were acquired with a microFlex™ mass spectrometer (Bruker Daltonik GmbH) in the positive mode. Bacterial Test Standard (Bruker Daltonik GmbH) was used for Instrument Calibration. Spectra were automatically interpreted by the database MBT Compass^®^ 4.1. A log (score) ≥ 1.7 was the threshold for the genus level identification and a log (score) of ≥ 2.0 was the threshold for the species level identification. Out of the original 76 isolates, 1 isolate did not grow and another isolate was classified as *Streptococcus* spp. at the genus level with MALDI score < 2.0, while the other 3 were identified as *Streptococcus dysgalactiae*. The remaining 71 strains were confirmed as *Strep. uberis* and were included in the study.

### 4.4. Molecular Characterization

DNA was extracted from the 71 confirmed *Strep. uberis* isolates following the protocol described by Cremonesi and collaborators [54]. After concentration and quality determination with a NanoDrop ND-1000, the DNA was amplified via mPCR to verify the presence of 4 housekeeping genes (*cpn60, gapC, sodA* and *tuf*) and to determine the occurrence of 6 genes related to virulence (*hasA, hasB, hasC, oppF, pauA, sua*), according to Calonzi and collaborators [19]. All the isolates were further characterized by a standard PCR assay to investigate *lbp* (lbp-FOR: 5′-GAGGCTGGCAACAAAGAACT-3′; lbp-REV: 5′-GCTTGTGCTTGGTTGTTTTG-3′). The in silico specificity was checked by using the BLAST software tool (https://blast.ncbi.nlm.nih.gov/Blast.cgi; accessed on 1 November 2019). The primers were synthesized by Thermo Fisher Scientific. The following cycling conditions were used 94 °C for 5 min, followed by 30 cycles of 94 °C for 1 min, 56 °C for 1 min and 72 °C for 1 min, and final extension of 72 °C for 7 min. The 25-μL reaction sample contained 12.5 μL of PCR Master Mix 2× (Thermo Fisher Scientific Waltham, MA, United States), 10.1 μL of Nuclease-free water, and 0.2 μL of each primer (100 μ*M*) and 2 μL of genomic DNA (5 ng/μL). Ten microliters of PCR product was electrophoresed on 2% agarose gel added with ethidium bromide (0.05 mg/mL; Sigma Aldrich, Milan, Italy). A 100 bp DNA ladder (Finnzymes, Espoo, Finland) was included in each gel. The results were visualized on an UV transilluminator (BioView Ltd., Nes Ziona, Israel). The positive control used in this study was *Strep. uberis* ATCC 9927 strain.

All the isolates were typed by RAPD-PCR analysis performed with primer ERIC1 (5’-ATGTAAGCTCCTGGGGATTCAC-3’). Amplification conditions, electrophoresis, and analysis of the amplification products were the same as those described by Schmitt-Van de Leemput and Zadoks [55].

### 4.5. Cluster Analysis

Grouping of the RAPD-PCR profiles was carried out using the BioNumeric 5.1 software package (Applied Maths, Sint-Martens-Latem, Belgium). The resulting dendrogram was created by the unweighted pair group method with arithmetic averages (UPGMA) cluster analysis; strains sharing the same number and the same size of PCR bands were considered genetically identical, while any relationship >90% and <100%, was defined as closely related. 

### 4.6. Antimicrobial Susceptibility Testing

The Minimum Inhibitory Concentration (MIC) of 14 antimicrobials was determined using the broth dilution test according to the procedure described in the Clinical and Laboratory Standards Institute (CLSI) guidelines VET015th edition [56]. MIC was evaluated with a customized commercial microdilution MIC system (Micronaut-S MIC Mastitis, Merlin Diagnostika, GmbH, Bornheim, Germany) used for routine laboratory testing of mastitis isolates. The MIC value of each isolate, expressed as μg/mL, was defined as the lowest concentration of the antimicrobial agent that completely inhibited the growth after the incubation period. *Streptococcus pneumoniae* ATCC 49619 was used as a quality control strain in each MIC batch, according to the reference values provided by CLSI VET08 guidelines, and a double-positive control was used for each plate. The antimicrobials were selected based on their activity against dairy cattle pathogens and on their registrations. The selected antimicrobials included 8 β-lactams (penicillin, ampicillin, oxacillin, amoxicillin plus clavulanate, cefazoline, ceftiofur, cefoperazone, cefquinome), enrofloxacin, erythromycin, florfenicol, lincomycin, tetracycline, and trimethoprim plus sulphametoxazole. Results were interpreted using available CLSI resistance breakpoints according to VET08 4th edition guidelines [25], the Comitè de l’Antibiogramme de la Sociètè Française de Microbiologie guidelines [28], the European Committee on Antimicrobial Susceptibility Testing guidelines [29] and the breakpoints reported in the literature [26,27], when specific standards were not established by any international recognized guidelines. The criteria used for the selections of the breakpoints were cattle, when available, human and other animal species breakpoints. The isolates with an intermediate MIC were classified as resistant. The MIC_50_ and MIC_90_ was calculated for each antimicrobial. MIC plate reading was performed manually and the last concentration of antimicrobial that did not show turbidity or a deposit of cells at the bottom of the well was recorded.

### 4.7. Statistical Analysis

Statistical analyses were performed using SPSS 27.0 Statistics for Windows (IBM, Armonk, NY, USA). Descriptive statistics of MIC values in different clusters were expressed as a mean ± SD. Normality of MIC value data distribution was assessed by the Shapiro–Wilk test. Since data were not normally distributed, the comparison of MIC values in different clusters was assessed using a nonparametric test (U Mann–Whitney) for 2 independent samples. *p* values < 0.05 were considered statistically significant.

## 5. Conclusions

In conclusion, *Strep. uberis* isolated from CM carried a combination of virulence genes, that might be linked to strains with a greater probability of causing clinical infections. The RAPD-PCR analysis carried out with ERIC1 showed a high frequency of closely related strains, whose occurrence might suggest their contagious nature. This study indicated that this molecular method can be a useful tool for investigating *Strep. uberis* mastitis. Even though the RAPD types obtained cannot be compared with those reported in other studies, this technique can provide insights about the epidemiology of *Strep. uberis* within the single dairy farm and help the understanding of clinical cases associated with this pathogen at the herd level. Some differences in the MIC values were found between clusters. AMR was widespread and multidrug resistant isolates were present but not prevalent. Our study underlines the need to consider MIC_50_ and MIC_90_ values when making farm management decisions because of the lack of breakpoints specific for mastitis pathogens. Genotypic and AMR results supported the concept that few *Strep uberis* isolates might be prevalent in the dairy herd; strategies aimed to control contagious mastitis may be useful to reduce *Strep. uberis* spreading within the farm. Surveillance data can be meaningful for practical management and helpful for the identification of the most appropriate antibiotic agents.

## Figures and Tables

**Figure 1 antibiotics-10-00644-f001:**
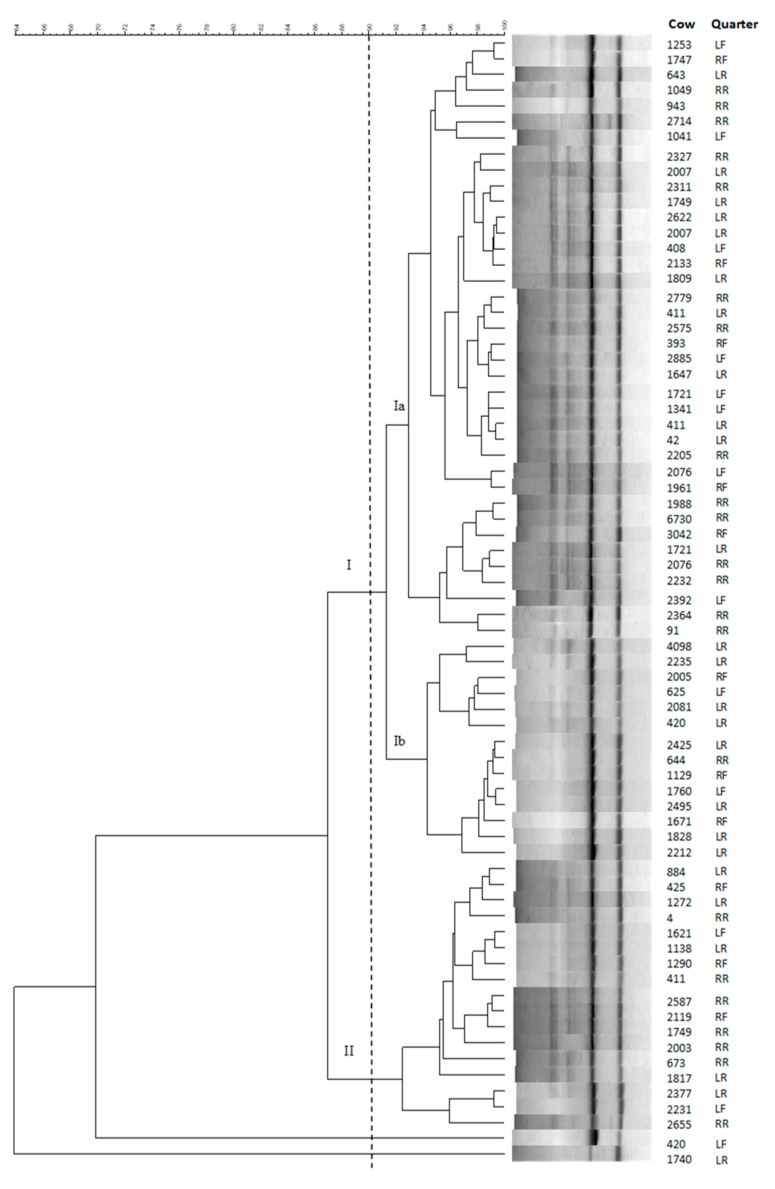
Unweighted pair group method with arithmetic averages (UPGMA)-based dendrogram derived from the RAPD-PCR profiles generated with primer ERIC1 of the 71 *Strep. uberis* strains isolated from 71 quarters of different cows.

**Table 1 antibiotics-10-00644-t001:** Antimicrobials tested, dilution range, breakpoints values, percentage of susceptible, intermediate and resistant *Strep. uberis* isolates, MIC inhibiting the growth of at least 50% (MIC_50_) and 90%(MIC_90_) of the 71 isolates analyzed.

Antimicrobials	Range (µg/mL)	Breakpoints (µg/mL) and Susceptibility						MIC_50_ (μg/mL)	MIC_90_ (μg/mL)	Reference
		S ^1^	[%]	I ^2^	[%]	R ^3^	[%]			
AMC ^4^	0.064/0.032–64/32	≤0.25/0.125	98.6	0.5/0.25	0	>1/0.5	1.4	0.25/0.125	0.25/0.125	[25]
AMP ^5^	0.016–16	≤0.25	98.6	0.5	0	>2	1.4	0.25	0.25	[25]
CEZ ^6^	0.125–32	≤2	98.6	4	0	>8	1.4	0.5	0.5	[25]
CPZ ^7^	0.125–16	≤2	97.2	4	1.4	>8	1.4	1	2	[26]
CEQ ^8^	0.125–8	≤2	98.6			>4	1.4	≤0.125	0.25	[27]
CEF ^9^	0.125–32	≤2	98.6	4	0	>8	1.4	0.5	1	[25]
ENRO ^10^	0.016–8	≤0.5	91.6	1–2	7	>4	1.4	0.5	0.5	[25]
ERY ^11^	0.125–8	≤0.25	91.6	0.5	2.8	>1	5.6	≤0.125	≤0.125	[25]
FLL ^12^	0.064–64	≤2	94.4	4	5.6	>8	0	2	2	[25]
LIN ^13^	1–8	≤2	7	4–8	11.3	>18	81.7	>8	>8	[28]
OXA ^14^	0.125–4	≤2	95.8			>4	4.2	2	2	[28]
PEN ^15^	0.0625–16	≤0.125	84.5	0.25–2	12.7	>4	2.8	0.125	0.25	[25]
TET ^16^	0.032–16	≤2	14.1	4	2.8	>8	83.1	>16	>16	[25]
T/S ^17^	0.016/0.304–32/608	≤1/19	98.6			>2/38	1.4	0.062/1.18	0.125/2.37	[29]

^1^ Susceptible, ^2^ Intermediate, ^3^ Resistant, ^4^ Amoxicillin/clavulanic acid, ^5^ Ampicillin, ^6^ Cefazolin, ^7^ Cefoperazone, ^8^ Cefquinome, ^9^ Ceftiofur, ^10^ Enrofloxacin, ^11^ Erythromycin, ^12^ Florfenicol, ^13^ Lincomycin, ^14^ Oxacillin, ^15^ Penicillin, ^16^ Tetracycline, ^17^ Trimethoprim/sulfamethoxazole.

**Table 2 antibiotics-10-00644-t002:** Distribution of the MIC_50_ and MIC_90_ for the 14 antimicrobial agents between the genotypic clusters, and cumulative percentage of *Strep. uberis* strains inhibited by their relative concentrations.

Antimicrobial Agents	Cluster I				Cluster II			
	MIC_50_		MIC_90_		MIC_50_		MIC_90_	
	(μg/mL)	[%]	(μg/mL)	[%]	(μg/mL)	[%]	(μg/mL)	[%]
AMC ^1^	0.25/0.125	98	0.25/0.125	98	0.125/0.064	53	0.25/0.125	100
AMP ^2^	0.25	98	0.25	98	0.125	53	0.25	100
CEZ ^3^	0.5	92	0.5	92	0.5	88	1	100
CPZ ^4^	1	73	2	96	1	65	2	100
CEQ ^5^	0.25	96	0.25	96	≤0.125	71	0.25	94
CEF ^6^	0.5	69	1	96	0.5	94	0.5	94
ENRO ^7^	0.5	88	1	94	0.5	100	0.5	100
ERY ^8^	≤0.125	90	≤0.125	90	≤0.125	94	≤0.125	94
FLL ^9^	2	94	2	94	2	94	2	94
LIN ^10^	>8	100	>8	100	>8	100	>8	100
OXA ^11^	2	96	2	96	2	94	2	94
PEN ^12^	0.125	81	0.25	96	0.125	94	0.125	94
TET ^13^	>16	100	>16	100	>16	100	>16	100
T/S ^14^	0.062/1.18	83	0.125/2.37	96	0.062/1.18	94	0.062/1.18	94

^1^ Amoxicillin/clavulanic acid, ^2^ Ampicillin, ^3^ Cefazolin, ^4^ Cefoperazone, ^5^ Cefquinome., ^6^ Ceftiofur, ^7^ Enrofloxacin, ^8^ Erythromycin, ^9^ Florfenicol, ^10^ Lincomycin, ^11^ Oxacillin, ^12^ Penicillin, ^13^ Tetracycline, ^14^ Trimethoprim/sulfamethoxazole.

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
