# Peer review of "Genotyping and Antimicrobial Susceptibility Profiling of *Streptococcus uberis* Isolated from a Clinical Bovine Mastitis Outbreak in a Dairy Farm"

_antibiotics, 2021, doi:10.3390/antibiotics10060644_

Round 1

Reviewer 1 Report

The manuscript presents some interesting aspects of S. uberis epidemiology in clinical mastitis cases in an Italian dairy farm.

However, some details in methodology and results presentation could be enhanced.

Figure 1:

- More information should be provided for the studied isolates, at least quarter and animal data.  

- Resistance data could also be added to the manuscript as character variable to assist further discussion.

- Most bands are faint; perhaps enabling bands highlight for better visualization  

Lines 161 – 163: a bit repetitive; is this an observation based on your results or a citation?

Lines 164 – 206: last paragraph of discussion section is too long; this could be separated in at least three paragraphs to ensure proper understating and discussion.

217: how many animals were evaluated? The 76 described below?

Lines 229 – 230: “76 isolates from quarter milk samples collected at the first CM case from 76 different cows” – does this means that you used only one isolate per animal? But did you have different isolations among quarters? If you had 2 quarters from one animal positive for S. uberis how did you select? Why not assess isolates from different quarters for genetic heterogeneity? considering the low number of isolates. This data should also be given in the dendrogram for proper understanding.

Line 231: which “routine biochemical identification” were used?

Line 240: “Bruker Compass software” – do you mean “MBT Compass®”?

Lines 242 – 243: what was the identification of the remaining 5 isolates?

Line 267: what were the tolerance and optimization parameters applied?

Lines 274 -298: It would be better to reorder the Antimicrobial Susceptibility Testing topic for proper understanding; give all the information about the customized MIC plate (lines 294-298, 291-293) and then results interpretation and breakpoints, perhaps as 287-290 and 280-286.  

Reviewer 2 Report

The manuscript describes molecular epidemiology of 71 isolates of Streptococcus uberis isolated from bovine clinical mastitis cases as well as susceptibility profile toward 14 different antimicrobials. The origin of isolated bacteria is a single farm monitored during specified time span.  

In the Introduction section authors offer key information regarding the significance of S. uberis in modern dairy production, molecular methods applied in research of streptococci and state of the art regarding susceptibility/resistance of S. uberis toward commonly used antimicrobials. The stated facts and cited literature give the rationale for the research.

In the Results section authors presented results in an unambiguous manner. The tables and figures are self explanatory and contain all necessary explanations.

In the Discussion section authors compared their own results with the results from comparable researches.

In the Material and method section authors describe applied methods with enough details.

Only minor suggestion is directed toward methods section and relies on positive/negative controls used in PCR protocols.

The final conclusions are in line with the achieved results.

Overall the manuscript is clearly written and easy to follow. Methods are well described.  Obtained results offer new facts and widen the current knowledge on the pathogen.

From the point of view of this reviewer the manuscript deserves to be published after aforementioned minor revision.

Author Response

Comments and Suggestions for Authors
Reviewer: 2
The manuscript describes molecular epidemiology of 71 isolates of Streptococcus uberis isolated from bovine clinical mastitis cases as well as susceptibility profile toward 14 different antimicrobials. The origin
of isolated bacteria is a single farm monitored during specified time span.
In the Introduction section authors offer key information regarding the significance of S. uberis in modern dairy production, molecular methods applied in research of streptococci and state of the art regarding susceptibility/resistance of S. uberis toward commonly used antimicrobials. The stated facts and cited literature give the rationale for the research.
In the Results section authors presented results in an unambiguous manner. The tables and figures are self explanatory and contain all necessary explanations.
In the Discussion section authors compared their own results with the results from comparable researches.
In the Material and method section authors describe applied methods with enough details.
Only minor suggestion is directed toward methods section and relies on positive/negative controls used in PCR protocols.
The final conclusions are in line with the achieved results.
Overall the manuscript is clearly written and easy to follow. Methods are well described. Obtained results offer new facts and widen the current knowledge on the pathogen.
From the point of view of this reviewer the manuscript deserves to be published after aforementioned minor revision.

AU: Thank you for the review and your valuable suggestion. At new line 273-274, the positive control used in this study (Streptococcus uberis ATCC 9927 strain) has been added, as suggested; for negative controls in PCR the authors used only bi-distilled water. The primers used were previously tested with different S.
aureus (DSMZ 19040, DSMZ 19041, DSMZ 19048, ATCC 25923, ATCC 19095), Str. agalactiae (ATCC BAA680), and other ATCC strains.
